# Enzymatic Crosslinked Hydrogels of Gelatin and Poly (Vinyl Alcohol) Loaded with Probiotic Bacteria as Oral Delivery System

**DOI:** 10.3390/pharmaceutics14122759

**Published:** 2022-12-09

**Authors:** Aldo F. Corona-Escalera, Ernesto Tinajero-Díaz, Rudy A. García-Reyes, Gabriel Luna-Bárcenas, Ali Seyfoddin, José Daniel Padilla-de la Rosa, Marisela González-Ávila, Zaira Y. García-Carvajal

**Affiliations:** 1Unidad de Biotecnología Médica y Farmacéutica, Centro de Investigación y Asistencia en Tecnología y Diseño del Estado de Jalisco, A.C. (CIATEJ), Guadalajara 44270, Mexico; 2Centro de Investigación y de Estudios Avanzados del IPN (CINVESTAV, Unidad Querétaro), Santiago de Querétaro 76230, Mexico; 3Drug Delivery Research Group, School of Science, Faculty of Health and Environmental Sciences, Auckland University of Technology, Auckland 1010, New Zealand; 4Tecnología Alimentaria, Biotecnología Industrial, Centro de Investigación y Asistencia en Tecnología y Diseño del Estado de Jalisco, A.C. (CIATEJ), Guadalajara 45019, Mexico

**Keywords:** hydrogel, gelatin, poly (vinyl alcohol), microbial transglutaminase, enzymatic, crosslinking, probiotic, survivability, lyophilization, gastrointestinal conditions

## Abstract

Probiotic bacteria are widely used to prepare pharmaceutical products and functional foods because they promote and sustain health. Nonetheless, probiotic viability is prone to decrease under gastrointestinal conditions. In this investigation, *Lactiplantibacillus plantarum* spp. CM-CNRG TB98 was entrapped in a gelatin–poly (vinyl alcohol) (Gel–PVA) hydrogel which was prepared by a “green” route using microbial transglutaminase (mTGase), which acts as a crosslinking agent. The hydrogel was fully characterized and its ability to entrap and protect *L. plantarum* from the lyophilization process and under simulated gastric and intestine conditions was explored. The Gel–PVA hydrogel showed a high probiotic loading efficiency (>90%) and survivability from the lyophilization process (91%) of the total bacteria entrapped. Under gastric conditions, no disintegration of the hydrogel was observed, keeping *L. plantarum* protected with a survival rate of >94%. While in the intestinal fluid the hydrogel is completely dissolved, helping to release probiotics. A Gel–PVA hydrogel is suitable for a probiotic oral administration system due to its physicochemical properties, lack of cytotoxicity, and the protection it offers *L. plantarum* under gastric conditions.

## 1. Introduction

Currently, probiotics are used to control and treat infections in the food and pharmacy industries [1,2]. The most important route for probiotic application is the oral route, which can be reached through diverse dosage forms including hydrogels, oral films, capsules, tablets, and lyophilized powders, etc. [3]. However, obtaining stable formulations of probiotics that can overcome physicochemical, pharmaceutical, and biological barriers to ensure and maximize their therapeutic efficacy and clinical applicability is challenging. Probiotic protection by encapsulation and use of surface-coating technologies enhances probiotic gastrointestinal tract (GIT) stability. Moreover, the encapsulated probiotics must remain sufficiently viable throughout the hydrogel manufacture, transport, and storage of the product and during passage through the adverse environment of the GIT [3,4].

Hydrogels are appropriate materials for encapsulating biological elements such as probiotic bacteria [5] due to their biocompatibility, high moisture content, softness, flexibility, and versatile fabrication [6].

Hydrogels can be fabricated from natural polymers (e.g., gelatin (Gel), pectin, chitosan, etc.) [7,8,9] or synthetic polymers (e.g., polyvinyl alcohol (PVA), Eudragit polymers, etc.), for probiotic encapsulation [10,11]. Although natural polymers are preferred to prepare hydrogels due to their low toxicity, eco-friendly properties, and low cost, they are mechanically weaker (having an inability to withstand loads) and more susceptible to degradation under physiological conditions than synthetic polymers, thus limiting their applications [12]. In order to improve the physicochemical and mechanical properties of hydrogels, binary and even ternary mixtures of natural and synthetic polymers have been made using eco-friendly routes for hydrogel preparation [13].

Gelatin (Gel) is a protein obtained from the hydrolysis of collagen. It is widely used to manufacture hydrogels for probiotic encapsulation [14]. Gel is a polypeptide composed of repeating triplets of alanine, glycine, and proline residues, which are responsible for gelatin’s typical triple-helical structure. It has excellent properties in terms of biodegradability, biocompatibility, and non-immunogenicity [15].

Poly(vinyl alcohol) (PVA) is a synthetic and biocompatible polymer that has been used in a wide range of industrial, commercial, medical, and food applications such as probiotic encapsulation [16]. PVA is a safe coating agent for pharmaceutical and dietary supplement products [17]. Additionally, properties such as high water content, bio-compatibility, swelling, and an elastic nature in a swollen state make PVA an excellent candidate for designing novel hydrogels [18].

An emerging approach for the in situ formation of hydrogels is based on enzyme-catalyzed crosslinking reactions due to the formation of complex architectures when using an eco-friendly process [19]. Microbial transglutaminase (mTGase) enzyme is an acyltransferase that catalyzes the amide–transferase reaction between the γ-amyl group of glutamine residue and the ε-amino group of lysine in proteins, modifying their chemical structure by intramolecular or intermolecular crosslinking favoring gel formation and thus improving the final use of the protein [20]. Previous works have demonstrated that gelatin’s mechanical and thermal properties can be improved by crosslinking gelatin molecules and even by crosslinking with polymers [21].

Designing oral administration systems for probiotics in various solid forms for their encapsulation, protection, and oral dosage while guaranteeing the minimum bioavailability requirements in the GIT is quite a challenge.

In this work, we focus on developing a new and straightforward method to enzymatically prepare probiotic-containing hydrogels for potential use as an oral delivery system for intestinal health in humans and animals. Gel–PVA-based hydrogels were enzymatically crosslinked by an mTGase enzyme to protect *L. plantarum* viability after lyophilization and under simulated gastrointestinal conditions. Hydrogel characterization was carried out in terms of swelling (%), porosity (%), and apparent density, and degradation under different conditions. Characterization was also carried out by using scanning electron microscopy (SEM), Fourier transform infrared spectroscopy (FTIR), thermogravimetric analyses (TGA), differential scanning calorimetry (DSC), and nuclear magnetic resonance (NMR). Additionally, the cytotoxicity was evaluated by using HT29 intestinal cells.

## 2. Materials and Experimental Section

### 2.1. Materials

*Lactiplantibacillus plantarum* BI-59.1 probiotic was obtained from the collection of the ex vivo digestion laboratory of the Center for Research and Assistance in Technology and Design of the State of Jalisco, A. C. (CIATEJ, A.C., Jalisco, Mexico). Gelatin from porcine skin (gel strength 300, type A) and poly (vinyl alcohol) (PVA) (89–98 kDa, 99+% hydrolyzed) were acquired from Sigma Aldrich (St. Louis, MO, USA). Microbial transglutaminase (mTGase, E.C.2.3.2.13) was purchased from BDF (Girona, Spain). Pepsin from porcine gastric mucosa and pancreatin from porcine pancreas were purchased from Sigma Aldrich (St. Louis, MO, USA). MRS agar and MRS broth were obtained from Difco (Detroit, USA). All other reagents were analytical grade. Double-distilled water was used throughout the experiments. Other solvents were used as received without further purification.

### 2.2. Probiotic Cultivation and Preparation

*L. plantarum* probiotics were activated in an MRS broth at 37 °C for 16 h and harvested by centrifugation at 5000× *g* for 5 min at 4 °C. The pellet was then washed with phosphate-buffered saline (PBS, pH 7.4) and collected by centrifugation, as was previously mentioned [22,23]. The recollected probiotics were resuspended in PBS and divided into bacteria for encapsulation and free bacteria for control.

### 2.3. Hydrogel Formation by Enzymatic Route

The preparation of the Gel–PVA hydrogels was as follows: Gel, PVA, and mTGase solutions were prepared individually. A 10 (wt%) solution of gelatin was prepared using deionized water magnetically stirred at 40 °C for 2 h. A 10 (wt%) solution of PVA was prepared by dissolving the polymer in deionized water at 80 °C under magnetic stirring for 2 h. A (10 wt%) solution of mTGase was prepared by dissolving the enzyme in distilled water [24,25]; this solution was centrifuged at 2760× *g* for 10 min. Afterwards, the enzyme solution was filtered using a 0.45 µm filter. The solution was then re-filtered using a 0.20 μm filter to guarantee sterilization (i.e., for cases when probiotics were incorporated into the hydrogel).

The polymer blend (Gel and PVA) solutions were prepared in a weight–polymer ratio of 1:1 under mechanical stirring at 40 °C for 2 h until homogeneous suspensions were obtained. The polymer blends underwent a pasteurization process (i.e., 60 °C for 20 min, followed by immediately being left to cool at 4 °C) and were used for probiotic entrapment [26,27].

Hydrogels were formed by adding amounts of the mTGase solution (10 U/g of Gel) into the Gel–PVA blend [25]. Afterward, the Gel–PVA–mTGase solutions were mechanically stirred at 40 °C for 5 min. They were then poured into molds of a cylinder-like shape (500 µg per mold). The molds were incubated at 40 °C for 40 min; subsequently, the enzyme was inactivated at 70 °C for at least 15 min [28]. Polymer samples were cooled down, kept at 4 °C overnight, and frozen at −80 °C for at least 12 h before lyophilization.

#### Probiotic Incorporation

After the pasteurization process, 9 Log CFU of *L. plantarum* cells per mL of pasteurized Gel–PVA polymer blend were added under mechanical stirring at 40 °C for 10 min. Hydrogels were subsequently formed as previously described (Section 2.3), bypassing enzymatic inactivation to prevent probiotic loss of viability.

### 2.4. Hydrogel Characterization

#### 2.4.1. Morphological Studies (SEM)

The macrostructure and pore size of the lyophilized hydrogels (with and without probiotics) were determined using scanning electron microscopy (SEM). Lyophilized hydrogels (with and without probiotics) were cut using liquid nitrogen. The dried samples were mounted on aluminum stubs with double-sided adhesive tape and coated with a thin layer of an Au–Pd cover using a Sputtering Plasma Sciences Inc. (0.3 kV voltage and 125 mTorr voltage, vacuum pressure) in a chamber under an argon atmosphere (SPI Supplies, Division of Structure Probe Inc., Westchester, NY, USA) and observed using a scanning electron microscope (Philips XL30 ESEM microscope, Amsterdam, The Netherlands).

#### 2.4.2. Nuclear Magnetic Resonance Spectroscopy (NMR)

^1^H spectra were recorded on a Bruker AMX-300 (Billerica, MA, USA) spectrometer at 25 °C operating at 300.1 and 75.5 MHz, respectively. Compounds were dissolved in deuterated water (D_2_O), and spectra were internally referenced to tetramethylsilane (TMS). About 10 mg of sample in 1 mL of solvent was used, and sixty-four scans were recorded for ^1^H-NMR. Pristine polymers and lyophilized hydrogels (without probiotics) were analyzed.

#### 2.4.3. FTIR-ATR Spectroscopy

Fourier transform infrared spectroscopy (FTIR) with an attenuated total reflectance (ATR) spectra in the range of 4000–450 cm^−1^ of the pristine polymers and lyophilized hydrogels (without probiotics) were obtained with a PerkinElmer spectrometer (Waltham, MA, USA), and eight scans were made with a resolution of 4 cm^−1^. The FTIR spectra were taken in transmittance mode.

#### 2.4.4. Thermogravimetric Analysis (TGA)

Thermogravimetric analysis (TGA) was performed on a Mettler Toledo TGA/DSC 1 Star System (Columbus, OH, USA) under a nitrogen flow of 20 mL/min^−1^ at a heating rate of 10 °C min^−1^ and within a temperature range of 30 to 600 °C. Pristine polymers and lyophilized hydrogels (without probiotics) were analyzed.

#### 2.4.5. Differential Scanning Calorimetry (DSC)

The reversible thermal behavior was examined by differential scanning calorimetry (DSC) using a PerkinElmer Pyris apparatus. Thermograms were registered from 4–6 mg samples at heating and cooling rates of 10 °C min^−1^ under a nitrogen flow of 20 mL/min^−1^. Indium and zinc were used as standards for temperature and enthalpy calibration. Pristine polymers and lyophilized hydrogels (without probiotics) were analyzed.

#### 2.4.6. X-ray Diffraction

The X-ray diffraction patterns of the hydrogels and raw materials were obtained using a RIGAKU Ultima-IV (Tokyo, Japan) diffractometer (40 kV, 30 mA) with Cu Kα1,2 (λ1 = 1.5406 Å, λ2 = 1.5443 Å) radiation. The X-ray diffraction patterns were collected with a scan rate of 4.2 degrees/min. Pristine polymers and lyophilized hydrogels (without probiotics) were analyzed.

#### 2.4.7. Water Absorption

The water absorption of the lyophilized hydrogels (with and without probiotics) was measured by gravimetric analysis. Dried samples with a fixed weight (~0.0350 mg) were immersed in distilled water at room temperature for 24 h. The excess water was removed from the free surface and weighed. The percentage of water absorption in distilled water was calculated as shown [29]:(1)$$\mathrm{Water}\;\mathrm{absorption}\;\left(\%\right)=\left(\frac{S-W}{W}\right)\times 100$$
where *W* is the sample weight in the dried state and *S* is the sample weight in the swollen state.

#### 2.4.8. Density and Porosity Measurement

Lyophilized hydrogels (with and without probiotics) were used for density and porosity measurements.

Density measurement was carried out using the solvent displacement method. Dried samples with cylindrical shapes were immersed in distilled water for 24 h at room temperature. Apparent density was calculated using the equation:(2)$$\rho =\frac{4m}{\pi d^{2}\;h}$$
where ρ is the apparent density $\left(g/{\mathrm{cm}}^{3}\right),$
*m* denotes the mass of the hydrogel after saturation with water, *d* is the diameter of the hydrogel after saturation with water, and *h* is the height of the hydrogel after saturation with water [29].

For porosity measurement, cylindrical dried hydrogels were immersed in absolute ethanol (EtOH) for 24 h and weighed after the excess of EtOH on the surface was blotted [30]. The porosity was calculated using the equation
(3)$$\mathrm{Porosity}\;\left(\%\right)=\frac{M_{h}-M_{d}}{\rho V}\times 100$$
where *M_d_* is defined as the weight of the hydrogel before immersion and $M_{h}$ indicates the weight of the hydrogel after it is immersed in ethanol. ρ and V represent the density of the ethanol and the volume of the sample, respectively.

#### 2.4.9. Hydrogels In Vitro Degradation in Simulated Gastrointestinal Fluids

The in vitro degradation in simulated gastrointestinal fluids was investigated by immersing lyophilized hydrogels (with and without probiotics) in distilled water (DW, pH 6.5 ± 0.1), simulated gastric fluid (SGF, pH 2.0 ± 0.1), and simulated intestinal fluid (SIF, pH 6.8 ± 0.1).

The simulated gastric fluid (SGF) was prepared by adding 2.0 g of NaCl in 70 mL of 1.0 N HCl and enough distilled water to obtain 1 L. The pH was adjusted to 2.0 ± 0.1 with 0.2 N HCl or 0.2 N NaOH. The solution was sterilized before adding 3.2 g of purified pepsin from porcine stomach mucosa [22,31]. The SIF was prepared by adding 6.8 g of monobasic potassium phosphate in 250 mL of distilled water followed by 77 mL of 0.2 N NaOH and enough distilled water to obtain 1 L. The pH was adjusted to 6.8 ± 0.1 with 0.2 N NaOH or 0.2 N HCl. The solution was then sterilized before adding 10 g of purified pancreatin and 1% bile salt [22,26,31].

The in vitro degradation test of the hydrogels was performed using the mass–loss index before and after being under distilled water, SGF, and SIF and was calculated from the average value of the mass change.

Dried hydrogels (~0.0350 mg) were immersed in 5 mL of each degradation medium (DW, SFG, and SIF) and placed in an incubator shaker with a shaking rate of 100 rpm at 37 °C for 2 h. Subsequently, the hydrogels were collected and dried in an oven at 60 °C for 24 h. The weight of the hydrogels was calculated before and after the degradation test. The experiment was performed in triplicates. The weight loss of all samples of hydrogel was calculated by the following equation [29,32]:(4)$$WL\left(\%\right)=\left(\frac{W_{0}-W_{f}}{W_{0}}\right)\times 100$$
where *WL* is the percentage of mass loss, $W_{0}$ is the initial mass of the samples before the degradation test, and $W_{f}$ is the final mass of the samples after the degradation test.

### 2.5. Biological Evaluation

#### 2.5.1. In Vitro Cytotoxicity: MTT Assay

Evaluation of the cytotoxicity of the Gel–PVA hydrogel was determined indirectly using the MTT technique [3-(4,5-Dimethyl-2-thiazolyl)-2,5-diphenyl-2H-tetrazolium bromide] according to the international standard ISO 10993-5:2009 and using a human colonic adenocarcinoma HT29-MTX (ECACC 12040401) cell line [33]. Prior to this analysis, extracts from the hydrogel were obtained by placing Gel–PVA hydrogel (UV-irradiated for 30 min) in DMEM (Sigma-Aldrich, St. Louis, MO, USA) medium without antibiotic and fetal bovine serum (FBS) and incubating the medium for 72 h at 37 °C. The hydrogel was then removed and the extract medium was diluted with DMEM and adjusted to 10% FBS to obtain the following extract concentrations: 12.5, 25, 50, 75, and 100%.

Cell culture: The HT29-MTX (ECACC 12040401) cells were cultured in DMEM supplemented with 10% inactivated FBS (Gibco, Waltham, MA, USA), penicillin (100 units/mL), and streptomycin (100 μg mL^−1^). Cells were incubated at 37 °C in a humidified atmosphere of 5% CO_2_. The medium was replaced every two days. After reaching 80% confluency, the cells were detached using 0.05% trypsin/1 mM ethylene diamine tetraacetic acid and assayed for viability using trypan blue exclusion assay (Sigma-Aldrich, St. Louis, MO, USA). Cells were seeded at a final density of 1 × 10^4^ cells/well and incubated for 24 h at 37 °C and with an atmosphere of 5% CO_2_. Afterwards, the cell culture medium was replaced with the extract medium at different concentrations [34].

Cytotoxicity of the hydrogel extract: Cells were exposed to the extract media of the hydrogels [33]. HT29-MTX cells in a complete DMEM medium were used as a negative control. Following this, 100 μL of extracts at different concentrations were added to a 96-well plate coated with HT29-MTX cells. The cells were cultivated with the extracts at 37 °C under a 5% CO_2_ atmosphere. After 24 h of exposure, the culture medium was removed and 100 μL of MTT was added at a concentration of 0.5 mg/mL in DMEM. This were incubated for 4 h at 37 °C and the appearance of blue formazan crystals was observed. Afterwards, 100 μL of dimethyl sulfoxide (Sigma-Aldrich, St. Louis, MO, USA) was added to each well to dissolve the formazan crystals. The absorbance was measured using a microplate spectrophotometer (Multiskan GO, Thermo Scientific, Waltham, MA, USA) at 570 nm. The following equation calculates the cell viability percentage [34]:(5)$$\mathrm{cell}\;\mathrm{viability}\;\left(\%\right)=\frac{I_{sample}}{I_{control}}\times 100$$
where $I_{sample}$ is the absorbance of the sample (cells incubated with extract media hydrogel), and $I_{control}$ is the absorbance of the control (cells incubated without extract media hydrogel).

#### 2.5.2. Probiotic Loading Efficiency (PLE)

The probiotic loading efficiency (PLE) was determined as the number of viable cells entrapped inside the Gel–PVA hydrogel divided by the initial amount of cells incorporated into the system. The PLE was expressed in Equation (6):(6)$$\mathrm{PLE}\;\left(\%\right)=\frac{N}{N_{0}}\times 100$$
where *N* is the log CFU mL^−1^ after entrapment and $N_{0}$ is the log CFU mL^−1^ initially incorporated [22].

The number of viable cells was determined by dispersing 0.5 g of probiotic-loaded hydrogels (before being frozen) in 4.5 mL of simulated intestinal fluid (SIF) to disintegrate the hydrogel and release the bacteria completely. The samples were placed in an incubator shaker with a shaking rate of 100 rpm at 37 °C for 3 h. Cells were counted in MRS agar using serial dilutions from the initial suspension and the drop plate technique. [35]. Samples then were incubated at 37 °C for 48 h. The viable cells were expressed as Log CFU per mL (log CFU mL^−1^).

#### 2.5.3. Probiotic Survival Determination after the Lyophilization Process

Lyophilized hydrogels (equivalent to 0.5 g of hydrated hydrogel) containing probiotics were dispersed in 5 mL of SIF and placed in an incubator shaker with a shaking rate of 100 rpm at 37 °C for 3 h. After incubation under aerobic conditions at 37 °C for 24 h, bacterial colonies were enumerated and the results were expressed as log CFU mL^−1^. Viable cells were then determined, as was previously mentioned in Section 2.5.2.

*L. plantarum* survivability was determined as follows [36]:(7)$$S\left(\%\right)=\frac{\mathrm{Probiotics}\;\mathrm{count}\;\mathrm{after}\;\mathrm{lyophiliozation}\;\mathrm{process}}{\mathrm{Probiotics}\;\mathrm{count}\;\mathrm{prior}\;\mathrm{to}\;\mathrm{lyophilization}\;\left(\mathrm{PLE}\right)}\times 100$$

#### 2.5.4. Hydrogel Behavior under Digestive Conditions

The viability of *L. plantarum* probiotics entrapped in the hydrogel was tested under simulated digestive conditions.

Lyophilized Gel–PVA hydrogels (the equivalent to 500 µg of hydrated hydrogel) loaded with *L. plantarum* were immersed in 5 mL of SGF and placed in an incubator shaker with a shaking rate of 100 rpm at 37 °C for 2 h. Afterwards, the samples were centrifuged (5000× *g* for 5 min at 4 °C) [26] and SGF was replaced with 5 mL of SIF. The sample then remained under the same incubation conditions for 3 h to allow complete disintegration of the Gel–PVA hydrogel and the release of probiotics.

For comparison, free *L. plantarum* cells (equivalent Log CFU content on the lyophilized Gel–PVA hydrogel) were evaluated under SGF and SIF.

As was previously mentioned, viable *L. plantarum* cells were determined (Section 2.5.2). After incubation under aerobic conditions at 37 °C for 24 h, bacterial colonies were enumerated and the results were presented as Log CFU mL^−1^.

### 2.6. Statistical Analysis

Data were reported as mean ± standard deviations of three independent experiments. A one-way analysis of variance was applied and, wherever appropriate, Tukey’s test was used to determine differences among the means (*p* < 0.05). Statistical analyses were performed with Statgraphics Centurion XVI program version 16.1.17 (Statgraphics Technologies, Inc., The Plains, USA).

## 3. Results and Discussion

Our studies aim to prepare and characterize new hydrogels and explore their application to encapsulating probiotics based on selected natural gelatin in combination with PVA by using an enzyme (mTGase) which acts as a crosslinking agent. Figure 1 displays the enzymatic pathway to synthesize the Gel–PVA hydrogel.

Microbial transglutaminase (mTGase, E.C.2.3.2.13) is able to catalyze the crosslinking of gelatin between the γ-carboxamide group of a glutamine residue and the ε-amino group of a lysine residue, leading to the formation of a covalent inter-molecular ε-(γ-glutamyl)lysine isopeptide bond which is resistant to physical and chemical degradation [37]. However, to our knowledge, there are no reports describing the preparation of Gel–PVA hydrogels using mTGase as a crosslinking agent and probiotic carrier.

### 3.1. Hydrogel Characterization

#### 3.1.1. Morphological Studies (SEM)

The morphology, macrostructure, and pore size of Gel–PVA lyophilized hydrogels were analyzed using scanning electron microscopy (SEM). Figure 1a,b shows SEM micrographs of hydrogels’ cross- and longitudinal-sections without probiotics. Meanwhile, Figure 1c shows the cross-section of the probiotic-loaded hydrogel.

The hydrogel presents an interconnected, crosslinked, highly macroporous macrostructure with well-formed microchannels and thin wall thickness, demonstrating an internal 3D network formation. Furthermore, the macroporous structures were characterized by having pores with diameters between 20–300 μm and heterogeneity in the macrostructure (Figure 1a–c).

The formation of the macroporous structure is due to the formation of ice crystals (large and small). The matter secreted or accumulated when frozen, which is subsequently lyophilized, results in a porous macrostructure with microchannels where the ice crystals reside. A reduction in the size of the microchannels is due to the increased difficulty for the ice crystals to form interconnected, crosslinked, and highly macroporous macrostructure [38,39,40].

Thangprasert et al. (2019) reported that the preparation of chemically (using glutaraldehyde) and physically crosslinked Gel–PVA hydrogels using the freeze–thaw method showed macrostructures and pore sizes similar to that of the hydrogels obtained by enzymatic and physical crosslinking (freeze–thaw) [41].

Figure 1c shows the presence of the probiotic bacteria entrapped in the wall of the macrostructure, confirming bacteria retention inside the hydrogels after the lyophilization process and indicating that enzymatic crosslinking is an excellent route for probiotic encapsulation.

#### 3.1.2. Nuclear Magnetic Resonance Spectroscopy (NMR)

To confirm that esterification reactions between Gel and PVA are mediated by the enzyme, we recorded ^1^H-NMR experiments in deuterated water. The ^1^H-NMR spectra of Gel, PVA, and the Gel–PVA hydrogel, respectively, are comparatively shown in Figure 2. Gel and PVA are D_2_O-soluble, but the Gel–PVA hydrogel is scarcely D_2_O-soluble. Therefore, to run the ^1^H-NMR analysis the sample was sonicated 48 h prior to collecting its spectrum. We observed a partial dissolution of the hydrogel, which slightly swelled, suggesting that crosslinking has occurred.

Interestingly, in the Gel–PVA hydrogel spectrum (Figure 2c) there are identified characteristic peaks from both precursors; for instance, at 1.65 ppm and 4.05 ppm peaks arose from the CH_2_ and HCOH hydrogens, respectively, of the PVA repeat units. In addition, the presence of aromatic residues at 7.24 ppm and the signals at 3.00 ppm and 0.87 ppm from lysine and methyl residues of valine, leucine, and isoleucine from gelatin in the hydrogel spectrum bring into evidence the coexistence of a crosslinked polymer [42,43]. Nonetheless, due to the complexity of the spectra in which overlapped peaks are found, NMR analysis is not conclusive enough to confirm an esterified product. Therefore, to further comprehend the supramolecular structure of the hydrogel, we performed FTIR experiments over a pristine sample of the hydrogel.

#### 3.1.3. FTIR-ATR Spectroscopy

The FTIR-ATR spectra of gelatin, PVA, and Gel–PVA hydrogel obtained over pristine samples are shown in Figure 3. The FTIR-ATR spectra show characteristic peaks of the precursor polymers; for instance, the spectrum of gelatin shows peaks at 3500 cm^−1^ and 3423 cm^−1^ due to ─NH stretching of secondary amide, a solid band for the C=O group appears at 1640 cm^−1^, and the ─NH bending appears at 1526 cm^−1^. Main absorptions of PVA functional groups are identified at 3280 cm^−1^ as a broad peak due to the vibration of free O─H (hydroxyl) bonds and the bending vibration of C─H bonds at 2915 cm^−1^.

The FTIR-ATR spectrum of the Gel–PVA hydrogel exhibits characteristic peaks of the precursors in the 3500 cm^−1^–2900 cm^−1^ range. Interestingly, at 1730 cm^−1^, a new sharp peak emerges due to the absorption of a carbonyl group C=O, the result of the formation of an esterified product, i.e., partial esterification of the free carboxylic groups of gelatin. Other authors have reported these observations, supporting our FTIR analysis of an esterification reaction between free carboxylic and hydroxyl groups in gelatin and PVA, respectively [44,45].

#### 3.1.4. Thermal Properties of the Gel–PVA Hydrogel

The thermal behavior of Gel–PVA hydrogels and pure polymers (Gel and PVA) was analyzed by TGA and DSC. Table 1 collects the thermal parameters and Figure 4 shows (a) thermogravimetry analysis (TGA) and (b) DSC thermograms.

TGA measures the change of weight of a sample as a function of a temperature profile, which helps to determine sample decomposition or the loss of solvent or water [34]. The thermal stability of the synthesized Gel–PVA hydrogel was examined by TGA under an inert atmosphere. The TGA traces were recorded in the 25–600 °C range for the hydrogel and the two precursors, Gel and PVA, respectively (Figure 4a). The decomposition of the hydrogel began at 250 °C and it took place through a single main step at a ^max^*T_d_* of 321 °C with a weight loss of about 60%, followed by a second minor step at a ^max^*T_d_* of 427 °C. The homopolymers started to decompose above 274 and 260 °C for the Gel and the PVA, respectively. The 5% weight reduction in the hydrogel at around 71 °C is attributed to moisture loss [42].

The thermal behavior of the Gel–PVA hydrogel was explored using DSC. The DSC traces registered at both heating and cooling for the hydrogel, and its precursors are displayed in Figure 4b. The characteristic events observed in this analysis are collected in Table 1.

The Gel–PVA hydrogel exhibits a large endothermic upward peak situated in the 50–75 °C region, which may reflect the evaporation of non-crystallizable water [46]. Interestingly, such an endothermic transition disappeared in the second heating cycle. This thermal behavior is directly attributed to its precursors, which exhibit a similar and characteristic behavior [47].

The glass transition temperature (Tg) for PVA was observed at 70 °C and for Gel at 82 °C. In the case of gelatin, the Tg is associated with the movement of amino acid blocks in the peptide chain of the triple helix. The Tg is slightly increased in gelatins with higher strength (Bloom value 300) [48]. The hydrogel Tg was found at 75 °C, decreased with respect to pristine dry Gel and increased with respect to dry PVA powder. This indicates movements inside amorphous regions in the polymer network of the hydrogel [48,49].

Moreover, the presence of mTGase favors crossover points in the peptide bonds of Gel triple helices [20] and the formation of hydrogen bonds with the Gel–PVA polymer chains (see FTIR analysis), modifying the mobility of the chains in the hydrogel three-dimensional network and changing their Tg [49].

TGA and DSC thermograms showed different thermal patterns of pristine Gel and PVA versus the hydrogels. The enzymatic crosslinking between Gel and PVA increases thermal stability, and these crosslinks provide probiotic protection in the gastric environment and delivery capability in the intestinal environment.

#### 3.1.5. X-ray Diffraction

The crystal structure of mTGase, Gel, PVA, and the Gel–PVA hydrogels (with and without probiotics) as characterized by X-ray diffraction (XRD) patterns are presented in Figure 5. The XRD pattern of the enzyme (mTGase) shows an amorphous pattern. The Gel pattern shows two typical characteristic diffraction peaks at 2θ ≈ 8.5°, related to the diameter of the triple helix, and at 22.5°, corresponding to the amorphous nature of gelatin [48,50].

The XRD pattern of PVA exhibits two well-defined peaks centered at approximately 2θ ≈ 11°, 20°, and 40°, indicating the semicrystalline nature of PVA, which contains crystalline and amorphous regions [40,51].

The XRD pattern of the hydrogel with probiotics exhibits diffraction peaks similar to the 2θ values of Gel and PVA, indicating that they have similar structural features. However, the intensities of the semicrystalline peaks of Gel and PVA decrease in the hydrogel, suggesting that the size of the crystalline domains is reduced in the presence of the enzyme. It is important to note that there was a significant decrease in the intensity of the peak at 8.5° in the hydrogel (characteristic in the Gel pattern), reflecting a reduction in the content of the triple helix structure and indicating a successful crosslinking among peptides of the Gel and enzyme, even in the presence of PVA. The XRD pattern of the loaded hydrogels did not exhibit any peaks, demonstrating that the incorporation of the probiotics favors the formation of the amorphous state inside the hydrogel.

Although there are no reports on the preparation of mTGase-crosslinked Gel–PVA hydrogels, Dong et al. reported on the preparation of hydrogels composed of gelatin-tin/cellulose nanocrystals (Gel-TG-CNC) cross-linked with mTGase, demonstrating that hydrogen bonds between CNCs and gelatin destroy the packing of the Gel structure and decrease the crystallinity of the composites [50].

#### 3.1.6. Water Absorption

The swelling behavior of a hydrogel plays an essential role in its practical applications [52]. Hydration capacity is an important issue in probiotic efficacy because probiotics are exposed to humidity during storage and in an aqueous environment during the GIT transit by oral administration [53,54].

Water absorption is essential in hydrogels delivering bioactive principles and interest molecules. This has a direct relationship to the release of the incorporated compounds and, in turn, is dependent on the natural chemistry of the polymers that constitute the hydrogel and the distribution of its pores [55]. Furthermore, absorbing up to thousands of times its dry weight in water [56] is one of the representative qualities of a hydrogel.

Figure 6 shows the water absorption capacity of Gel–PVA hydrogels with and without probiotics at one point and in contact with distilled water. The hydrogels exhibited 1300% w/w of water absorbed (retained) in their network with no noticeable change, even when containing probiotics. This is because the hydrogel’s structure is very similar between the samples, characterized by macroporous and microchannels along the system (see SEM images).

On the other hand, Gel–PVA hydrogels have adequate water absorption capacity owing to the hydrophilic functional groups (i.e., amino/–NH_2_, carboxyl/–COOH, and hydroxyl/–OH groups) on the polymer chain with very significant hydrogen bonding [57].

The deprotonation of the –COOH groups in the gelatin and the protonation of the –OH groups in PVAs played an essential role when the pH values were adjusted in the aqueous medium where the enzymatic crosslinking took place, as when they are close to the isoelectric point of gelatin there is an anion/cation relationship and electrostatic repulsion between the polymeric chains is favored. This induces an “expansion” in the macrostructure of the hydrogels and therefore the water absorption profile (aqueous media) will be different in each type of hydrogel [58].

Long et al. reported that water uptake for a gelatin system enzymatically cross-linked with mTGase was ~1200% due to the high hydrophobicity of gelatin [59], which is similar to the capacity obtained in our Gel–PVA hydrogel. This is probably due to the esterification of the carboxylic group. However, the amino groups remain present, playing an important role due to their hydrophilic nature. The result obtained by our research group is higher than that reported by other authors regarding gelatin and PVA systems. For example, Pal et al. reported a system capable of absorbing ~260% of its dry weight [60].

#### 3.1.7. Porosity and Apparent Density

Porosities and bulk density values of Gel–PVA hydrogels are shown in Figure 7a,b, respectively. The hydrogel samples showed statistical differences (*p* < 0.05) in porosity and apparent density values. The Gel–PVA hydrogel without probiotics presented a porosity percentage of 20.72 ± 0.62% while the hydrogel loaded with probiotics showed porosity of 15.61 ± 0.54% (Figure 7a). It was observed that the porosity depends on the volume of the pores in the hydrogel. The loaded hydrogels’ porosity decreased as probiotics impeded the formation of large ice crystals and therefore smaller pores were formed. SEM images of the loaded hydrogel revealed pore diameters between ~30–150 μm. Other research groups obtained similar results. For example, Long et al. reported the formation of Gel-based sponges crosslinked with glutaminase with a pore size of 100 µm [59], and Thangprasert et al. prepared PVA–Gel hydrogels at different polymer ratios using the freeze–thaw method with a subsequent crosslink with glutaraldehyde demonstrating a pore–size distribution between 30–120 µm [41].

Apparent or bulk density is defined as the dry hydrogel mass per unit volume (g/cm^3^). The bulk density is an indicator that helps determine how compact the dry hydrogel is and is the relationship between the volume of the solutes and pores it has. The lower the bulk density value, the more compact the hydrogel will be. The apparent density represents the relationship between solids and pore space in the macrostructure and therefore varies with the solute content and its molecular weight. Bulk density can also provide information on the hydrogel’s porosity, permeability, diffusion, and mechanical properties [29].

The Gel–PVA hydrogels without probiotics showed apparent density values of 0.58 ± 0.02 g/cm^3^, while the loaded hydrogel exhibited 0.63 ± 0.01 g/cm^3^ (Figure 7b). The apparent density values of the loaded hydrogels have increased noticeably. The formation of small ice crystals (or eventually of amorphous ice) makes the matter accumulated between adjacent microchannels less densely packed and amorphous (XRD analysis supports these results, Figure 5). Variations in the density values could be ascribed to the amorphous or crystallinity state of the hydrogels [40].

A relationship can be observed between the % of porosity (displacement of the solvent) and the apparent density given that, with more pore space, the apparent density decreases, which is consistent with what has been previously reported by other authors [29,61].

#### 3.1.8. In Vitro Degradation of Hydrogel under Different Conditions

The degradation behavior of Gel–PVA hydrogels (with and without probiotics) in water and gastrointestinal media limits their application as a delivery system for oral probiotic bacteria because it compromises the viability of probiotics. Knowing the susceptibility of hydrogels to hydrolytic and proteolytic degradation (digestive enzymes) provides relevant information about their structural stability and biological performance [62].

A simple and quick method to measure the degradation of hydrogels consists of determining % mass loss (degradation) after hydrogels were soaked in three different media after 24 h: distilled water (DW, pH 6.5 ± 0.1), simulated gastric fluid (SGF, pH 2.0 ± 0.1), and simulated intestinal fluid (SIF, pH 6.8 ± 0.1). A reduction in mass was observed in all cases after the degradation test. No statistical differences were found between the Gel–PVA hydrogel with or without *L. plantarum* cells. (*p* > 0.05). The mass loss (%) values at different fluids showed the following order: SIF (41.72 ± 4.60%) > SGF (30 ± 0.14%) > DW (12.32% ± 0.46). The highest degradation occurred in the SIF (see Figure 8).

This hydrolytic degradation is mainly associated with water’s solvation between the remaining polymer chains. According to Ceylan et al., the mass loss in the system is due to the Gel as PVA is not affected by hydrolytic cleavage [63].

Intestinal fluids contain the enzymes and salts characteristic of most digestive systems. One of the main requirements to be met by oral systems intended to deliver principles of interest and probiotic bacteria is to protect the latter from gastrointestinal conditions, mainly from the stomach’s acidic environment. For this reason, the Gel–PVA hydrogels were subjected to a resistance test, finding significant differences between the hydrogel in simulated gastric fluid (SGF pH 2.0 ± 0.01) and simulated intestinal fluid (SIF, pH 6.8 ± 0.01).

The degradation under gastric and intestinal conditions is due to the solvation of water with the remaining polymer chains and the proteolytic action of enzymes [64]. However, the Gel–PVA hydrogel presents less mass loss in SGF due to the presence of PVA as, in addition to the esterification reaction with gelatin, it is not degraded by the enzymes in the human body. The degradation of PVA is mainly due to the facile particle disintegration and dissolution over time [65] which occur in the SIF.

During production and storage Gel–PVA hydrogels are exposed to moisture, which promotes water absorption and hydrolytic degradation. If the hydrogels were ingested, they would be subjected to digestion and intestinal transit where they would undergo a transition that would result in their breakdown and decomposition as a result of the intestinal microenvironment (stomach acidity, presence of enzymes, bile salts, etc.) [66].

The in vitro degradation assays demonstrate the structural stability of Gel–PVA hydrogel under DW and SGF. However, the pH, ionic strength, temperature stimuli, and SIF composition significantly affected the hydrogel’s swelling and disintegration. For this reason, SIF was chosen over PBS, as was reported by other authors [36], to completely disintegrate the Gel–PVA hydrogel and allow the complete release of entrapped probiotic bacteria which is required to calculate PLE, probiotic survival to the lyophilization process, and hydrogel behavior under digestive conditions.

### 3.2. Biological Evaluation

#### 3.2.1. Cytotoxicity

The objective of this test was to evaluate if there is a cytotoxic response to Gel–PVA hydrogels using different dilute extract media on HT29-MTX cells–intestinal epithelial cells from the human colon–after an incubation time of 24 h. Cytotoxicity is one of the most important properties for developing oral probiotics delivery systems for intestinal health.

Figure 9 shows the values of the cell viability percentage for decreasing concentrations of Gel–PVA hydrogel extracts. The % cell viability at different concentration extracts showed the following order: 75% > 100:50:25% > 12.5%. It can be observed how the % viability exceeds 100% for all concentrations except for the 12.5% concentration (90%). The cell viability of the 100, 50, and 25% extracts did not show statistical differences (*p* < 0.05) after 24 h incubation. The extract concentration at 75% after 24 h reaches cell viability higher than 100%.

According to the ISO 10993-5 standard, a percentage of viability >70% using this technique is considered non-cytotoxic [33]. The Gel–PVA hydrogel is therefore considered non-cytotoxic for HT29-MTX cells. Comparable cell viability values were reported by our research group using Gel/PVA/Chitosan hydrogels and HT29-MTX cells. [34,67].

Cell viability of the Gel–PVA hydrogels was greater than 80%, indicating cytocompatibility and suggesting the hydrogels’ potential to be used as an oral delivery system for probiotics.

#### 3.2.2. Probiotic Loading Efficiency and Survival Probiotic after Lyophilization Process

Embedding is often mentioned to protect bacteria against environmental factors, processing conditions, storage, and GIT targeting [68]. The incorporation of probiotics into the Gel–PVA mixture was approximately 9.01 Log CFU mL^−1^. Following the crosslinking process and after being cooled at 4 °C for 24 h, the probiotic cells in the Gel–PVA hydrogel were, on average, 8.42 ± 0.07 Log CFU mL^−1^, representing a PLE of 93.44 ± 0.72% (Figure 10a). This is similar to the results reported by other authors [69,70] and higher than the minimum viable dose recommended for probiotic products consumed by humans, which is 6 Log CFU mL^−1^ (dashed black line) [71,72].

#### 3.2.3. Probiotic Survival to the Lyophilization Process

The viability of *L. plantarum* after lyophilization is shown in Figure 10b. Before lyophilization, the viable cells were around 8.42 ± 0.04 Log CFU mL^−1^. The viable cells after the lyophilization process decreased to 7.66 ± 0.05 Log CFU mL^−1^, i.e., 91.00 ± 0.64%, which concurs with the results previously reported by other authors [69,73]. This indicates that Gel–PVA hydrogels protect *L. plantarum* cells from the freeze-drying process, surpassing the minimum viable dose (dashed black line) recommended even though no traditional cryoprotectant such as sucrose or skim milk was used to prevent cell loss [74,75]. Nevertheless, the cryoprotective action could be linked to PVA, which has previously been reported as a cryoprotectant [76].

#### 3.2.4. Hydrogel Behavior under Digestive Conditions

Oral administration of probiotics is considerably more comfortable and safe. Probiotics must pass through the GIT from the mouth to the stomach to the small intestine until they reach the colon (their functional site) and colonize the intestinal mucosa in competition with native gut bacteria [77]. The viability of probiotics administered by this route is compromised by the harmful conditions of the GIT [68,77]. Therefore, the composition, physicochemical properties, and morphology of the probiotic carrier systems can influence the viability of probiotics during the pass-through of the GIT [78,79].

This test aimed to evaluate the potential of Gel–PVA hydrogels as protective delivery systems of probiotics under two simulated digestive conditions: the stomach and small intestine.

Figure 11 shows the survivability of free and entrapped *L. plantarum* cells under SGF (2 h) and SIF (3 h) conditions. The hydrogels had fast water absorbency in both fluids (SGF and SIF) due to the high porosity and interconnection among some pores within the polymer network (SEM images). As was previously mentioned in Section 3.1.7, due to the characteristics of SIF it was chosen to completely disaggregate the Gel–PVA hydrogel and allow the release of probiotic bacteria after being under SGF.

Free *L. plantarum* cells showed a survival rate after exposure to SGF at 37 °C during 2 h of 0% (initial value from 7.66 Log CFU mL^−1^ to 0 Log CFU mL^−1^), demonstrating that probiotics need to be protected from acidic conditions. This is consistent with the previous findings that *L. plantarum* BI-59.1 does not survive at pH 2.0 [80].

On the other hand, when the free bacteria were exposed to SIF the viable bacteria-rate values did not present statistical differences in the CFU values at the beginning and after 3 h of exposure to SIF at 37 °C (from 7.66 Log CFU mL^−1^ to 7.65 Log CFU mL^−1^), indicating that SIF does not have any adverse effect on this strain. This finding reinforces the election of SIF as the disintegration medium for Gel–PVA hydrogel as it will not modify the viable CFU.

In comparison, when the Gel–PVA hydrogels loaded with *L. plantarum* were exposed to SGF at 37 °C for 2 h and subsequently to SIF at 37 °C for 3 h, allowing the complete release of probiotic bacteria, the cell viability value observed was 7.20 ± 0.10 Log CFU mL^−1^, i.e., a reduction of 0.58 CFU Log CFU mL^−1^ compared to the initial value (7.66 ± 0.05 Log CFU mL^−1^). This represents a survival rate of 94.72 ± 1.31%.

The results show that the enzymatically crosslinked Gel–PVA hydrogel protects *L. plantarum* from gastric conditions and allows viable cells to release into the intestine.

## 4. Conclusions

This work demonstrated the manufacturing processes’ capacity for preparing a hydrogel based on Gel and PVA for possible applications in intestinal health. Probiotic-loaded Gel–PVA hydrogels were prepared by enzymatic crosslinking followed by subsequent lyophilization. The hydrogels showed dense, reticulated, macroporous, and interconnected structures containing bacteria, which were observed by SEM images. Furthermore, the hydrogels showed structural integrity due to the formation of interlocking networks. The enzymatic crosslinking process led to chemical modifications between Gel–PVA confirmed by FTIR and ^1^H-MNR. The TGA and DSC values indicate that the hydrogels are stable at body temperature. Additionally, the Tg values suggest movements within amorphous regions of the hydrogel polymeric network, which is reflected in the diffractograms (XRD). The hydrogels presented a high water-uptake capacity due to their porosity, with the charged hydrogels being more porous. Gel–PVA hydrogels were found to be biocompatible and non-toxic to HT29-MTX cells. *L. plantarum* cells were not affected by the enzymatic crosslinking, and the hydrogel protected them from the freeze-drying process, allowing the incorporation of adequate concentrations of probiotic bacteria.

In addition, the Gel–PVA hydrogel protected the probiotics from the stomach’s acidic conditions and allowed the probiotics’ release and survival in the simulated intestinal fluid containing bile salts. The unique characteristics of these hydrogels open a new field of application in targeted oral administration by using these gastro-resistant hydrogels as oral targeting delivery systems for probiotics and other bioactive for intestinal health.

## Data Availability

The data presented are available on request from the corresponding author.

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
