# Peer review of "Enzymatic Crosslinked Hydrogels of Gelatin and Poly (Vinyl Alcohol) Loaded with Probiotic Bacteria as Oral Delivery System"

_pharmaceutics, 2022, doi:10.3390/pharmaceutics14122759_

Round 1
Reviewer 1 Report
The manuscript presents the encapsulation of the probiotic strain L. plantarum in gelatin-poly (vinyl alcohol) hydrogel. The main novelty is the use of transglutaminase as crosslinking agent in order to obtain the gelatin-PVA hydrogel. It is an interesting article but needs to correct some issues before publication.
Abstract
Lines 24-25 this information does not need to be included in the abstract.
The results related to the protection against simulated gastrointestinal conditions should be included in the abstract
Introduction
The introduction is a bit long. Some aspects are, in my opinion, superfluous. For instance, lines 47-52 or lines 62-67.
Line 78-79 in probiotic delivery porosity can be adverse for a good protection. Nutrient transit is not a critical issue in this kind of application.
Material and methods
Line 107 -please indicate E.C.2.3.2.13
Line 121 – Pectin? Should be gelatin, no?
Line 125 - please use g instead of rpm
Line 132-137 is really needed to include 2 references for this information? Other examples as such could be given throughout the manuscript; it has to many references (99!) for an original work.
Line 142-143 – not clear
Line 245-246 – please explain how the extracts were obtained. 100% is the hydrogel itself?
Line 248-250 – this information is superfluous
Lines 261-263. Extracts were diluted but the concentration is 100%? Please explain.
Line 265 - MX
Lines 288-289 and 311-312- no need to repeat. Same as above 2.5.2
Results and discussion
Lines 327-347 – this information could be included in the introduction.
Line 370-372 - This may be the case for growing cells not for to be used as oral delivery system (non-growing cells)
Line 484 - only one reference seems to be related with probiotics and it is a review. Please cite the original work with probiotics that supports the sentence
Line 485-486- please rephrase. How can DSC and TGA analysis improve the stability and integrity?
Legend figure 6 and 7 – The bars indicate standard deviation?
Line 562-563 – how similar were the results to works cited? Please expand
Lines 624-657 – I have several concerns related to the use of SIF as disintegrating agent to estimate bacteria entrapped or released. How to be sure that all bacteria are released since there in no 100% disintegration? How was the disintegration compared to PBS? Some probiotic bacteria my suffer from detrimental effects of pancreatin and/ or bile salts, in this could PBS be used?
Lines 738-741- Any idea of survivability of the free cells?
Conclusions
Lines 798-799- I think that what should be said is the enzymatic cross-linking did not affect the probiotic bacteria viability
Author Response
Comment 1: The manuscript presents the encapsulation of the probiotic strain L. plantarum in gelatin-poly (vinyl alcohol) hydrogel. The main novelty is the use of transglutaminase as crosslinking agent in order to obtain the gelatin-PVA hydrogel. It is an interesting article but needs to correct some issues before publication.
Reply: Thank you for reading this extensive manuscript. Thanks for the comments. The manuscript was restructured. Please check the new version.
Comment 2: Abstract. Lines 24-25 this information does not need to be included in the abstract. The results related to the protection against simulated gastrointestinal conditions should be included in the abstract
Reply: Probiotic bacteria are widely used to prepare pharmaceutical products and functional foods because they promote and sustain health. Nonetheless, probiotic viability is prone to decrease under gastrointestinal conditions. In this investigation, Lactiplantibacillus plantarum spp CM-CNRG TB98 was entrapped in a gelatin-poly (vinyl alcohol) (Gel-PVA) hydrogel, which was prepared by a “green” route using microbial transglutaminase (mTGase), acting as a crosslinking agent. The hydrogel was fully characterized, and its ability to entrap and protect L. plantarum from the lyophilization process and under simulated gastric and intestine conditions was explored. The Gel-PVA hydrogel showed a high probiotic loading efficiency (> 90 %) and survivability to the lyophilization process (91 %) of the total bacteria entrapped. Under gastric conditions, no disintegration of the hydrogel was observed, keeping L. plantarum protected with a survival rate of > 94 %.. While in the intestinal fluid, the hydrogel is completely dissolved, which helps release probiotics. Gel-PVA hydrogel is suitable for a probiotic oral administration system due to its physicochemical properties, no cytotoxicity, and protection of L. plantarum under gastric conditions.
Comment 3: The introduction is a bit long. Some aspects are, in my opinion, superfluous. For instance, lines 47-52 or lines 62-67
Reply: lines 47-52 and 62-67 were deleted
Comment 4: Lines 78-79 in probiotic delivery porosity can be adverse for a good protection. Nutrient transit is not a critical issue in this kind of application.
Reply: Thank you for the comment. Lines 77-80 were deleted
Material and methods
Comment 5: Line 107 -please indicate E.C.2.3.2.13
Reply: The change was done
Comment 6: Line 121 – Pectin? Should be gelatin, no?
Reply: Thank you, the word pectin was changed to gelatin in the manuscript.
Comment 7: Line 125 - please use g instead of rpm
Reply: The change was made in the manuscript.
Comment 8: Line 132-137 is really needed to include 2 references for this information? Other examples as such could be given throughout the manuscript; it has to many references (99!) for an original work.
Reply: Thank you, the bibliography was reduced.
Comment 9: Line 142-143 – not clear
Reply: Thank you, these lines were rewritten.
After the pasteurization process, 9 Log CFU of L. plantarum cells per mL of pasteurized Gel-PVA polymer blend were added under mechanical stirring at 40 °C for 10 min. Subsequently, hydrogels were formed as previously described (section 2.3), bypassing enzymatic inactivation to prevent probiotic loss of viability.
Comment 10: Line 245-246 – please explain how the extracts were obtained. 100% is the hydrogel itself?
Reply: Thank you, this part was changed.
The evaluation of the cytotoxicity of Gel-PVA hydrogel was determined indirectly by the MTT technique [3-(4,5-Dimethyl-2-thiazolyl)-2,5-diphenyl-2H-tetrazolium bro-mide] according to the international standard ISO 10993-5:2009 using a human colonic adenocarcinoma HT29-MTX (ECACC 12040401) cell line [34]. Before this analysis, extracts from the hydrogel were obtained by placing Gel-PVA hydrogel (UV-irradiated for 30 min) in DMEM medium without antibiotic and fetal bovine serum (FBS) and incubated for 72 h at 37 °C. Then, the hydrogel was removed, and the extract media was diluted with DMEM and adjusted to 10% FBS to obtain the following extract concentrations: 12.5, 25, 50, 75, and 100%.
Comment 11: Line 248-250 – this information is superfluous
Reply: Thank you for the comment. Lines 248-250 were deleted
Comment 12: Lines 261-263. Extracts were diluted but the concentration is 100%? Please explain.
Reply: Extracts from the hydrogel were obtained by placing Gel-PVA hydrogel (UV-irradiated for 30 min) in DMEM medium without antibiotic and fetal bovine serum (FBS) and incubated for 72 h at 37 °C. Then, the hydrogel was removed, and the extract media was diluted with DMEM and adjusted to 10% FBS to obtain the following extract concentrations: 12.5, 25, 50, 75, and 100%.
In this case, the dilution is only necessary to achieve 12.5, 25, 50, and 75 % concentration.
Comment 13: Line 265 - MX
Reply: The extra MX was deleted
Comment 14: Lines 288-289 and 311-312- no need to repeat. Same as above 2.5.2
Reply: The text was changed
Results and discussion
Comment 15: Lines 327-347 – this information could be included in the introduction.
Reply: Some lines were deleted and added in the introduction.
Comment 16: Line 370-372 - This may be the case for growing cells not for to be used as oral delivery system (non-growing cells)
Reply: Thank you, lines 370-372 were deleted.
Comment 17: Line 484 - only one reference seems to be related with probiotics and it is a review. Please cite the original work with probiotics that supports the sentence
Reply: The change was made
Comment 18: Line 485-486- please rephrase. How can DSC and TGA analysis improve the stability and integrity?
Reply: The phrase was modified.
TGA and DSC thermograms showed different thermal patterns of pristine Gel, PVA, versus the hydrogels. The enzymatic crosslinking between Gel and PVA increases thermal stability, and these crosslinks provide probiotic protection in the gastric environment and delivery capability in the intestinal environment.
Comment 19: Legend figure 6 and 7 – The bars indicate standard deviation?
Reply: Legends were changed.
Comment 20: Line 562-563 – how similar were the results to works cited? Please expand
Reply: The phrase was modified. The reference 75, Miao et al. was deleted and changed by Long, et al.
SEM images of the loaded hydrogel revealed pore diameters between ~30-150 μm.
Other research groups obtained similar results, for example, Long et al. reported the formation of Gel-based sponges crosslinked with glutaminase with a pore size of 100 m [62], and Thangprasert et al. prepared PVA-Gel hydrogels at different polymer ratios by freeze-thaw with a subsequent crosslink with glutaraldehyde showing a pore size distribution between 30-120 m [41].
Comment 21: Lines 624-657 – I have several concerns related to the use of SIF as disintegrating agent to estimate bacteria entrapped or released. How to be sure that all bacteria are released since there in no 100% disintegration? How was the disintegration compared to PBS? Some probiotic bacteria my suffer from detrimental effects of pancreatin and/ or bile salts, in this could PBS be used?
Reply: In a previous trial, PBS was tested to disintegrate the hydrogel. However, the hydrogel remained intact even after more than 12 hours, so the bacteria were not completely released, and the viable count could not be considered appropriate. For this reason and by observing the results of the hydrogel degradation test (Figure 8), it was decided to replace the PBS with SIF. For this, the viability test of free cells in SIF was previously carried out to ensure that the viability was not diminished when L. Plantarum cells were in contact with the simulated fluid (Figure 11). As the viability of free cells under SIF was not affected, we proceeded to use SIF as a disintegrating agent. Under SIF, the hydrogel was entirely disintegrated, allowing the complete release of bacteria.
It’s possible that some probiotic bacteria may suffer some damage under SIF. In that case, PBS could be used, as long as this can disintegrate the matrix and allow complete bacteria release.
Comment 22: Lines 738-741- Any idea of survivability of the free cells?
Reply: Lyophilization of free cells was bypassed due to bacteria viability after the lyophilization process without incorporating a cryoprotectant is less than 40 %
https://doi.org/10.1016/j.cryobiol.2022.01.003
https://doi.org/10.1016/j.cryobiol.2022.01.003
Conclusions
Comment 23: Lines 798-799- I think that what should be said is the enzymatic crosslinking did not affect the probiotic bacteria viability
Reply: Thank you, this part was rewritten.

Reviewer 2 Report
· Some little mistakes:
Line 43: by instead of from (?)
Line 49: their instead of its
Line 86: their instead of its
Line 109: MRS agar, and MRS broth WERE (instead of was)
Line 635: parenthesis
Line 763: the instead of de
· Lines 57-59: could the authors provide more details on the compromised mechanical properties of hydrogels made of natural polymers?
· Line 121: did the authors mean <gelatin> where it is written “A solution of PECTIN was prepared”?
· The hydrogel production process is repeated twice: the first on lines 129-139 and the second on lines 140-148, where the initial addition of probiotics is also described. It seems quite redundant. To avoid this problem you could embed paragraph 2.3.1 (Probiotic incorporation) in paragraph 2.3 naming it Hydrogel formation by the enzymatic route and probiotic incorporation. Another consideration: in the case of the hydrogel loaded with probiotics, did the authors bypass the enzymatic inactivation?
· In the apparent density equation line 205 and in the porosity equation line 213 it is necessary to specify the units of measurement for each parameter mentioned.
· About in vitro digestion to evaluate hydrogel degradation: authors tested only SGF and SIF, separately. Has no combination test been performed?? (hydrogel prior digested in SGF and then in SIF, as described to test probiotic survival to digestion)
· Lines 285- 290: why did the authors test the probiotic encapsulation efficiency (or the survival to lyophilization, as written at lines 292-299) by testing the viability after 3 h in SIF? is it a way to free bacteria from hydrogels for counting? it's not very clear as written in the methods section (clear if you also read the discussion)
· In line 292 what does it mean fresh weight?
· In equation (7) what does it mean (EE) ?
· Lines: 322-348: this text is suitable for section 1. Introduction.
· Lines 527-528: no equilibrium over time can be deduced from figure 6
· Just a curiosity: authors have rightly described porosity as an important parameter to consider to understand if probiotics (or some other component) are trapped. Have other tests been done to see if probiotics are actually being withheld over time?
· Line 640: what do you mean for physical erosion?
· How did the authors justify that 75% Gel-PVA hydrogel extracts give better results than other percentages in terms of cell viability?
· In Figure 1c, it may be helpful to clearly demonstrate the presence of the LP. Either by providing a larger magnification photograph or by showing a photograph at the same magnification of a hydrogel sample without the addition of LP
Author Response
Comments and suggestions for authors
Reviewer 2
Thank you for reading this extensive manuscript. Thanks for the comments. The manuscript was restructured. Please check the new version.
Comment 1: Line 43: by instead of from (?)
Reply: The change was made
Comment 2: Line 49: their instead of its
Reply: The change was made
Comment 2: Line 86: their instead of its
Reply: The change was made
Comment 3: Line 109: MRS agar, and MRS broth WERE (instead of was)
Reply: Thank you, the change was made
Comment 4: Line 635: parenthesis
Reply: The parenthesis was added
Comment 5: Line 763: the instead of de
Reply: The changed was made
Comment 6: Lines 57-59: could the authors provide more details on the compromised mechanical properties of hydrogels made of natural polymers?
Reply: More details were added
Although natural polymers are preferred to prepare hydrogels due to their low toxicity, eco-friendly properties, and low cost, they are mechanically weaker (inability to withstand loads) and more susceptible to degradation under physiological conditions than synthetic polymers, limiting their applications [12].
Comment 7: Line 121: did the authors mean < Gelatin> where it is written “A solution of PECTIN was prepared”?
Reply: Thank you, the change was made.
Comment 8: The hydrogel production process is repeated twice: the first on lines 129-139 and the second on lines 140-148, where the initial addition of probiotics is also described. It seems quite redundant. To avoid this problem you could embed paragraph 2.3.1 (Probiotic incorporation) in paragraph 2.3 naming it Hydrogel formation by the enzymatic route and probiotic incorporation. Another consideration: in the case of the hydrogel loaded with probiotics, did the authors bypass the enzymatic inactivation?
Reply: Thank you, this part has been modified in the manuscript
Comment 9: In the apparent density equation line 205 and in the porosity equation line 213 it is necessary to specify the units of measurement for each parameter mentioned.
Reply: The units has been added in the manuscript
Comment 10: About in vitro digestion to evaluate hydrogel degradation: authors tested only SGF and SIF, separately. Has no combination test been performed?? (hydrogel prior digested in SGF and then in SIF, as described to test probiotic survival to digestion)
Reply: The objective of the degradation test was to determine the hydrogel weight loss of mass in each fluid, trying to understand the effect of different stimuli (pH, fluid composition). Therefore, the test was done in SGF and SIF separately and using hydrogel without probiotics. In the SGF the hydrogel resists the hydrolysis, but in the SIF, the hydrogels were disintegrated entirely.
The goal of the digestion test was to determine if the hydrogel protected probiotics from harmful GI environments even if the hydrogels disintegrated in the SIF. The hydrogel was first immersed in the SGF and subsequently in the SIF for this determination.
Comment 11: Lines 285- 290: why did the authors test the probiotic encapsulation efficiency (or the survival to lyophilization, as written at lines 292-299) by testing the viability after 3 h in SIF? is it a way to free bacteria from hydrogels for counting? it's not very clear as written in the methods section (clear if you also read the discussion)
Reply: In a previous trial, PBS was tested to disintegrate the hydrogel. However, the hydrogel remained intact even after more than 12 hours, so the bacteria were not completely released, and the viable count could not be considered appropriate. For this reason and by observing the results of the hydrogel degradation test (Figure 8), it was decided to replace the PBS with SIF. For this, the viability test of free cells in SIF was previously carried out to ensure that the viability was not diminished when L. Plantarum cells were in contact with the simulated fluid (Figure 11). As the viability of free cells under SIF was not affected, we proceeded to use SIF as a disintegrating agent. Under SIF, the hydrogel was entirely disintegrated, allowing the complete release of bacteria.
This part was rewritten for better comprehension.
Comment 12:In line 292 what does it mean fresh weight?
Reply: It refers to the weight of hydrogel before the lyophilization process
Comment 13: In equation (7) what does it mean (EE) ?
Reply: Thank you, EE was replaced for PLE. EE was an alternative way to name PLE
Comment 14: Lines: 322-348: this text is suitable for section 1. Introduction.
Reply: Some of the lines were deleted and added in the introduction.
Comment 15:Lines 527-528: no equilibrium over time can be deduced from figure 6
Reply: Thank you, these lines were deleted.
Comment 16: Just a curiosity: authors have rightly described porosity as an important parameter to consider to understand if probiotics (or some other component) are trapped. Have other tests been done to see if probiotics are actually being withheld over time?
Reply: Thank you, no other tests were performed to ensure probiotics were withheld over time since it was out of the scope of this particular work.
Comment 17: Line 640: what do you mean for physical erosion?
Reply: The words “physical erosion” were changed to facile particle disintegration
The degradation of PVA is mainly due to facile particle disintegration and dissolution over time [84], which occur in the SIF.
Comment 18: How did the authors justify that 75% Gel-PVA hydrogel extracts give better results than other percentages in terms of cell viability?
Reply:
The ISO 10993-5 guideline recommends three cytotoxicity test methods: direct and indirect contact and extract dilution. The extract test evaluates the cytotoxicity of any leachable byproducts from the macerated material. In our experiments, the cytotoxicity evaluation of the hydrogel was done using extract dilutions and the MTT assay. Even though the results obtained demonstrated that there is no cytotoxicity, it is observed that the 75% extract shows an increase in cell proliferation. To answer the reviewer's question with certainty, specific biological assays must be made to elucidate the high proliferation rate at that particular extract concentration, which is not in the scope of this research work.
Some studies show that Gelatin [https://doi.org/10.1002/jbm.b.30128, https://doi.org/10.1016/j.jff.2012.10.017; https://doi.org/10.1016/j.procbio.2021.08.002; https://doi.org/10.1177/0883911511423563] and PVA [https://doi.org/10.1016/j.heliyon.2021.e06182; https://doi.org/10.1111/gtc.12843 promotes the proliferation of various cell lines at specific concentrations due to different biological mechanisms.
However, we suggested the possibility that the 75% extract favors the conditions that allow greater HT29 cell proliferation because it contains optimal polymer concentrations (Gel: PVA) that favors cell proliferation modifying some signal pathways in the HT29MX cells. Tang et al, demonstrated that supplementing the cell medium with 1 mg/mL PVA for the expansion of human pluripotent stem cells (hPSC) resulted in an increase in cell density in the culture plate [doi: 10.1111/cpr.13112]. Khan, et al. showed that when they added 0.5% gelatin into cell media containing adipose-derived mesenchymal stem cells (Ad-MSCs) cell proliferation was enhanced.
Comment 19: In Figure 1c, it may be helpful to clearly demonstrate the presence of the LP. Either by providing a larger magnification photograph or by showing a photograph at the same magnification of a hydrogel sample without the addition of LP
Reply: Thank you, there is no larger magnification of Gel-PVA hydrogel without or with LP

Round 2
Reviewer 1 Report
The authors have addressed all the questions raised.
There are some typos that can be corrected during final formatting and proofing of the document:
-lines 52-53;
line 286;
lines 341-343;
Line 578;
Line 753 repeated information (significant difference)
Reviewer 2 Report
Authors fulfill all previous requested revisions.